# Protective Effects of Eicosapentaenoic Acid Plus Hydroxytyrosol Supplementation Against White Adipose Tissue Abnormalities in Mice Fed a High-Fat Diet

**DOI:** 10.3390/molecules25194433

**Published:** 2020-09-27

**Authors:** Paola Illesca, Rodrigo Valenzuela, Alejandra Espinosa, Francisca Echeverría, Sandra Soto-Alarcon, Cristian Campos, Alicia Rodriguez, Romina Vargas, Thea Magrone, Luis A. Videla

**Affiliations:** 1Laboratory of Studies of Metabolic Diseases Related to Nutrition. Faculty of Biochemistry. University of Litoral, Santa Fe 3000, Argentina; pillesca@fbcb.unl.edu.ar; 2Nutrition Department, Faculty of Medicine, University of Chile, Santiago 8380000, Chile; franciscaecheverria@uchile.cl (F.E.); sandra.soto.alarcon@gmail.com (S.S.A.); 3Medical Technology Department, Faculty of Medicine, University of Chile, Santiago 8380000, Chile; ealejand@uchile.cl (A.E.); cristian.campos.19@gmail.com (C.C.); 4Department of Food Science and Chemical Technology, Faculty of Chemical and Pharmaceutical Sciences, University of Chile, Santiago 8380494, Chile; arodrigm@uchile.cl; 5Molecular and Clinical Pharmacology Program, Institute of Biomedical Science, Faculty of Medicine, University of Chile, Santiago 8380000, Chile; rominavargas@med.uchile.cl (R.V.); lvidela1944@gmail.com (L.A.V.); 6Department of Basic Medical Sciences, Neuroscience and Sensory Organs, University of Bari, 70124 Bari, Italy; thea.magrone@uniba.it

**Keywords:** high-fat diet, white adipose tissue, adipocyte hypertrophy, metabolic dysfunction, eicosapentaenoic acid, hydroxytyrosol

## Abstract

Objective: Obesity induced by high-fat diet (HFD) elicits white adipose tissue dysfunction. In this study, we have hypothesized that the metabolic modulator eicosapentaenoic acid (EPA) combined with the antioxidant hydroxytyrosol (HT) attenuates HFD-induced white adipose tissue (WAT) alterations. Methods: C57BL/6J mice were administered with a HFD (60% fat, 20% protein, 20% carbohydrates) or control diet (CD; 10% fat, 20% protein, 70% carbohydrates), with or without EPA (50 mg/kg/day), HT (5 mg/kg/day), or both for 12 weeks. Determinations in WAT include morphological parameters, EPA and docosahexaenoic acid content in phospholipids (gas chromatography), lipogenesis, oxidative stress (OS) and inflammation markers, and gene expression and activities of transcription factors, such as sterol regulatory element-binding protein-1c (SREBP-1c), peroxisome proliferator-activated receptor-gamma (PPAR-γ), nuclear factor kappa-light-chain-enhancer of activated B cells (NF-κB) (p65 subunit) and nuclear factor erythroid 2-related factor 2 (Nrf2) (quantitative polymerase chain reaction and enzyme linked immunosorbent assay). Results: HFD led to WAT hypertrophy in relation to PPAR-γ downregulation. WAT metabolic dysfunction was characterized by upregulation of lipogenic SREBP-1c system, mitochondrial energy metabolism depression, loss of the antioxidant Nrf2 signaling with OS enhancement, n-3 long-chain polyunsaturated fatty acids depletion and activation of the pro-inflammatory NF-κB system. EPA and HT co-supplementation diminished HFD-dependent effects additively, reaching values close or similar to controls. Conclusion: Data presented strengthen the importance of combined protocols such as EPA plus HT to attenuate metabolic-inflammatory states triggered by obesity.

## 1. Introduction

Overweight and obesity have been defined by the World Health Organization (WHO) as the abnormal or excessive fat accumulation that may impair health [1]. It currently represents a worrying public health problem, since from 1975–1980 to the present, the global prevalence of overweight and obesity has doubled, reaching a third of the world′s population with this condition [2,3,4]. Obesity increases the risk of chronic noncommunicable diseases (NCDs) and triggers skeletal muscle disorders, sleep apnea and poor mental health [1], with development of non-alcoholic fatty liver disease (NAFLD) [5]. The fat accumulation that characterizes obesity promotes white adipose tissue (WAT) expansion, thus altering its normal functions. These include hypertrophy contributing to development of oxidative stress, the activation of low-grade inflammatory response and dysregulation of lipolysis accompanied by insulin resistance (IR) [6]. The prevention of obesity and associated syndromes has been assessed by the supplementation with different compounds, including natural products. Among them, the use of the n-3 long-chain polyunsaturated fatty acids (n-3 LCPUFAs), eicosapentaenoic acid (C20:5n-3, EPA) and docosahexaenoic acid (C22:6n-3, DHA) revealed effective outcomes not only related to lipid metabolism disorders, but also to cardiovascular diseases and inflammatory states [7]. EPA is biosynthesized from the essential precursor α-linolenic acid ((C18:3n-3), ALA) through desaturation and elongation reactions occurring primarily in the liver, with actions being exerted on signal transduction pathways and gene transcription [8]. These include downregulation of lipogenic sterol regulatory element-binding protein-1c (SREBP-1c), upregulation of peroxisome proliferator-activated receptor-α (PPAR-α) favoring fatty acid oxidation (FAO) and promotion of inflammation resolution via synthesis of specific pro-resolving mediators (SPMs) [9]. Besides n-3 LCPUFAs, antioxidant compounds also have positive results in NCDs, as is the case of the polyphenol hydroxytyrosol (HT) present in olive oil and red wine. HT has biological properties beyond its potent antioxidant capacity inhibiting oxidative stress [10], such as the modulation of lipid metabolism and inflammatory responses [11]. Both EPA and HT can activate transcription factors that modify the gene expression of downstream lipid, antioxidant and inflammatory mediators [11,12,13].

Currently, preclinical studies point to the advantage of using combined protocols over monotherapies in the prevention or resolution of metabolic disorders [14,15]. In fact, the co-supplementation protocols are more effective than the single ones, considering that the former exhibit the possibility to exert additive or potentiated responses [14,16]. In this respect, high-fat diet (HFD)-induced liver steatosis and pro-inflammatory status in mice were attenuated by combined EPA and HT administration in an additive manner [17,18]. The underlying mechanisms comprise the regulation of SREBP-1c and PPAR-α controlling lipogenesis and FAO respectively, as well as the content of the SPMs resolvin-1 and 2 abrogating inflammation [17,18]. According to these considerations, the aim of this study was to test the hypothesis that the co-administration of EPA and HT potentiates the effects of the separate supplementation with EPA or HT on adiposity enhancement and WAT dysfunction in HFD-fed mice.

## 2. Material and Methods

### 2.1. Animals, Diets and Experimental Procedures

All animal procedures in this study were in strict adherence to the Guide for the Care and Use of Laboratory Animals (National Academy of Sciences, NIH Publication 6–23, revised 1985) and were approved by the Bioethics Committee for Research in Animals, Faculty of Medicine, University of Chile (CBA # 0580 FMUCH). Weaning male C57BL/6J mice weighing 12–14 g (Bioterio Central, ICBM, Faculty of Medicine, University of Chile) were randomly assigned to each experimental group and allowed free access to a control diet (CD) or HFD. The composition of CD (expressed as % total calories) was 10% fat, 20% protein and 70% carbohydrate, with a caloric value of 3.85 kcal/g and free of EPA and DHA. The composition of the HFD was 60% fat, 20% protein and 20% carbohydrate, with a caloric value of 5.24 kcal/g and free of EPA and DHA (Research Diet INC, Rodent Diet, Product data D12450K and 12492, USA). Fatty acid composition of CD and HFD were previously published [19]. Animals received water ad libitum and were housed on a 12 h light/dark cycle from day 1 to 84 (12 weeks). As appropriate, the treated groups received orally through gavage, EPA and/or HT. EPA, isolated from fish oil (Golden Omega S.A., Chile) as triacylglycerols (TAG) (50% EPA, 5% DHA and 5% of other n-3 FAs, 15% saturated fatty acid (SFA) principally palmitic acid and 25% monounsaturated fatty acids (MUFAs) principally oleic acid). EPA was administered at 50 mg/kg/day dosage. HT (ela-Vida™, DSM Nutritional Products Company, Netherlands) was given at doses of 5 mg/kg/day. Control groups received isovolumetric amounts of saline orally, conforming eight experimental groups, namely: (a) CD + saline (control), (b) CD + EPA, (c) CD + HT, (d) CD + EPA + HT, (e) HFD + saline, (f) HFD + EPA, (g) HFD + HT and (h) HFD + EPA + HT. The doses of EPA and HT used in this study were in agreement with previous investigations: EPA (50 mg/kg) and HT (5 mg/kg), both supplied separately or in combination, and showed protective effects against HFD-induced steatosis [20]. In addition, HT at the doses mentioned above proved to prevent adipose tissue dysfunction induced by HFD [14]. Weekly controls of body weight (BW) and diet intake were performed through the whole period. At the end of the 12th week, animals were fasted (6–8 h). Under isoflurane anesthesia, subcutaneous, perirenal and epididymal fat pads were removed, weighed and immediately assayed or frozen in liquid nitrogen as appropriate. Epididymal white adipose tissue (EWAT) dissection was performed carefully to avoid contamination with subcutaneous adipose tissue. Blood samples were collected, and serum was obtained after centrifugation (1000× *g*, 10 min, 4 °C) for later analysis of adipokines. The animals were euthanized by cardiac puncture. EWAT was selected to perform the evaluations since it is metabolically representative of visceral adipose tissue in rodents and is the largest and most accessible fat pad.

### 2.2. Serum Adipokines Profile

ELISA kits were used for the assessment of serum levels of adiponectin (mg/L) (Cayman Chemical Company, number 10007620, Ann Arbor, MI, USA) and leptin (mg/mL) (Abcam Inc., number ab100718, Toronto, ON, Canada).

### 2.3. Morphological Parameters

Adipocytes from EWAT were isolated according to the method of Rodbell [21], as previously described [13]. The morphological parameters of adipocytes (size, mean diameter and cell volume) were determined according to Di Girolamo et al. [22]. The number of cells in the fat pad was estimated by measuring the lipid content by the method of Folch [23]. The lipid content of the average fat cell and the total number of cells in the fat pad was calculated as recently described in detail [13].

### 2.4. Fatty Acid Composition of Phospholipids

The methodology used to assess phospholipid fatty acid composition was described elsewhere [13,24]. Briefly, the total lipids from EWAT were extracted with chloroform/methanol (2:1 *v/v*) according to the method proposed by Bligh and Dyer [25]. Then, the phospholipids were isolated from the lipid extract by thin-layer chromatography performed on silica gel plates. The derivatization of phospholipids was carried out with boron trifluoride to produce fatty acid methyl esters (FAME), which were subsequently separated and quantified by gas–liquid chromatography.

### 2.5. Oxidative Stress Indicators and Non-Enzymatic Defenses

EWAT thiobarbituric acid reactive substances (TBARS) and F-2 isoprostanes contents were determined by colorimetric assays according to the manufacturer’s instructions (Cayman Chemical Company, Ann Arbor, MI, USA). Protein carbonyls concentration was determined by a fluorometric assay (Cayman Chemical Company, Ann Arbor, MI, USA) after adjusting the total protein concentration to 7.5 mg/mL per sample. Reduced glutathione (GSH) and glutathione disulfide (GSSG) contents in EWAT were assessed with an enzymatic recycling method described by Rahman et al. [26].

### 2.6. Activities of Antioxidant and Lipogenic Enzymes

Samples of EWAT were homogenized in three volumes of phosphate buffer (30 mM), pH 7.4 containing EDTA (1 mM) and sucrose (250 mM) for determination of antioxidant enzyme activities. The homogenate was subjected to a first centrifugation (1200× *g*, 10 min, 4 °C) and one supernatant aliquot was used to evaluate superoxide dismutase (SOD) (Cayman Chemical Co., Ann Arbor, MI, USA) and catalase (CAT) activities [27]. Subsequently, the remaining supernatants were centrifuged at 100,000× *g* for 60 min at 4 °C to perform glutathione peroxidase (GPX) and glutathione reductase (GR) assays [27]. On the other hand, EWAT samples were homogenized in three volumes of buffer, pH 7, containing KH_2_PO_4_ (9 mM), K_2_HPO_4_ (85 mM), dithiothreitol (DTT) (1 mM) and KHCO_3_ (70 mM) and centrifuged (1800× *g*, 10 min, 4 °C). The fat of the upper layers were discarded, the supernatants were again centrifuged at 100,000× *g* for 60 min at 4 °C and the aqueous fractions were used to study Acetyl-CoA carboxylase (ACC), fatty acid synthase (FAS) and the glucose-6-phosphate dehydrogenase (G6PD) activities as previously described [12,14]. To assay malic enzyme (ME) activity, EWAT homogenates were centrifuged for 10 min at 40,000× *g* at 4 °C, then the ME activity was evaluated in the resulting supernatant following the method performed by Wise and Ball [28]. In addition, the activity of lipoprotein lipase (LPL) (enzyme associated with the tissue incorporation of fatty acids) was measured in an epididymal fat pad according to Llobera et al. [29]. The protein content in enzyme extracts was determined by the Bradford assay (Bio-Rad reagent, California, USA).

### 2.7. SREBP-1c, PPAR-γ, NF-ΚΒ and Nrf2 DNA-Binding Activity

Nuclear extracts from EWAT were obtained using a commercial extraction kit (Cayman Chemical Company number 78782-99-7, Ann Arbor, MI, USA). SREBP-1c, PPAR-γ, nuclear factor-kB (NF-κB) (p65 subunit) and nuclear factor E2-related factor 2 (Nrf2) DNA-binding activities were assessed with a commercial ELISA kit (Cayman Chemical Company, number 10010854. SREBP-1c, number 10006855 by PPAR-γ, number 10007889 by NF-κB (p65) and number 600590 by Nrf2, Ann Arbor, MI, USA) and according to the manufacturer’s instructions. Values were expressed as percentage of SREBP-1c, PPAR-γ, NF-κB (p65 subunit) and Nrf2 DNA-binding with respect to a positive control provided by the respective ELISA kit.

### 2.8. Quantitative Real-Time PCR

Total RNA was isolated from EWAT samples using Trizol (Invitrogen, Carlsbad, CA, USA), according to the supplier’s protocols. Purified RNA (2 μg) was then treated with DNAase I (DNA free kit; Ambion, Austin, TX, USA) and used to generate first-strand cDNA with moloney murine leukemia virus reverse transcriptase (Invitrogen, Carlsbad, CA, USA), utilizing random hexamers (Invitrogen, Carlsbad, CA, USA) and deoxynucleotide triphosphate (dNTP) mix (Bioline, London, United Kingdom), according to the manufacturer’s protocol. The resultant cDNA was amplified with specific primers for mice in a total volume of 20 µL. Gene-specific primer sequences used are shown in Table 1. Primers were optimized to yield 95%–100% of reaction efficiency with PCR products by development in agarose gel to verify the correct amplification length. The quality of the DNA extracted was checked by agarose gel (1%) electrophoresis according to Lee et al. [30]. Briefly, samples were mixed with Gel loading dye (0.25% bromophenol blue, 0.25% xylene cyanol and 30% glycerol). Electrophoresis was run at 75 Volt for 20 min. DNA was stained using GelRed^®^ Nucleic Acid Gel Stain (Biotum, Fremont, CA, USA). Real-Time PCR was performed in a Stratagen Mx3000P System using Brilliant II SYBR^®^ Green quantitative real time PCR Master Mix (Agilent Technologies, La Jolla, CA, USA) following the manufacturer’s recommendations (Applied Biosystems, Foster City, CA, USA). All the expression levels of target genes under study were normalized by the expression of β-actin as an internal control (Applied Biosystems, Foster City, CA, USA). Fold changes between groups were calculated by the 2-(ΔΔCt) method, as established by Pfaffl [31] and Livak and Schmittgen [32].

### 2.9. Statistical Analysis

Statistical analysis was performed with GraphPad Prism 6.0 software (GraphPad Prism, Inc., San Diego, USA). The values shown represent the mean ± standard deviation (SD) for 8–10 animals per each experimental group. Evaluation of normality data distribution was performed using the Shapiro–Wilk test. Assessment of the statistical significance of differences between mean values was performed by two-way analysis of variance (ANOVA) and the Tukey’s test. A *p* < 0.05 was considered significant.

## 3. Results

### 3.1. HFD-Induced Obesity, Adiposity and WAT Hypertrophy are Decreased by EPA and HT Co-Administration

The body weight (BW) of mice at the beginning of the study was similar. After 12 weeks, all groups fed with CD, with or without supplementation, exhibited comparable final BW. HFD led to an increase in BW gain of 65% (*p* < 0.05), compared with CD groups, which is related with to augmented adiposity (Table 2A, group e). In fact, all fat pads evaluated, namely, subcutaneous, perirenal and epididymal, in mice fed the HFD showed significant weight increases (Table 2B), therefore the total mass of WAT increased considerably (198%; *p* < 0.05) over the CD group. Dietary intake was comparable in all experimental groups, but energy consumption was higher in mice subjected to the HFD with or without supplementation over CD groups (Data not shown). EPA and EPA + HT supplementation diminished BW gain produced by the HFD by 30% and 37% respectively (*p* < 0.05), accompanied with a decrease in fat pads and total WAT of 34% and 32% (*p* < 0.05), respectively (Table 2B). HT administration was not able to prevent the increased BW observed in HDF-fed mice; however, it reduced the localized and total adiposity (total WAT) by 28% (Table 2B). Specific and morphological parameters were evaluated in the EWAT (Table 2C). The expansion of epididymal fat pad of HFD-fed mice exhibited a lower number of cells per gram of WAT, adipocytes of increased size with augmented TAG content compared with the control group, indicating hypertrophy. The supplementation with EPA, HT or both significantly decreased the hypertrophy observed in WAT of the HFD-fed mice, improving all the parameters evaluated without observing differences between treatments and without reaching control values (Table 2C). Levels of serum adipokines are shown in the Table 2D. Serum adiponectin and leptin were not modified by supplementation when mice were fed the CD. However, the HFD significantly decreased serum adiponectin levels (39%; *p* < 0.05) while increasing those of leptin (136%; *p* < 0.05). Supplementation with EPA, HT or EPA + HT improved the profile of serum adipokines in HFD-fed mice, showing a similar effect independently of the treatment, although the values did not reach those observed in CD-fed mice.

### 3.2. HFD-Induced Changes in Fatty Acid Composition in EWAT Phospholipids are Improved by EPA and HT Co-Supplementation

Comparable content of total saturated fatty acids (SFA), monounsaturated fatty acids (MUFA) and polyunsaturated fatty acids (PUFA) were found in EWAT phospholipids of all groups receiving CD. HFD-fed mice exhibit an increase of total SFA and decrease of total PUFA (Table 3). Regarding the EWAT palmitate levels, comparable values were observed in all groups fed with CD, a significant increment in HFD saline group, and a decrease caused by the administration of EPA or HT alone, with the total recovery towards CD values when supplemented with EPA+HT (Table 3). Moreover, HFD led to a severe reduction in EPA content (81%; *p* < 0.05) with consequent decrease in DHA content (72%; *p* < 0.05) in EWAT phospholipids compared to CD-fed animals (Table 3). The high content of EPA in the adipose tissue of the animals that received the supplementation evidence the effectiveness of the treatment, increasing by 2.86-fold and 3-fold EPA levels in CD groups supplemented with EPA and EPA+HT, respectively. Moreover, the supplementation with EPA to HFD-mice (EPA and EPA+HT groups) restored the depleted levels of EPA in EWAT phospholipids reaching control levels. The incorporation of EPA into EWAT also improved the content of DHA in this tissue, increasing DHA levels in CD animals, with normalization in HFD mice achieving levels observed in CD group (Table 3). It should be noted that in the HFD-fed group that received HT exclusively, the levels of EPA and DHA were also increased in EWAT phospholipids, confirming previous results [18,20]. Surprisingly, the effect of the supplements given together, improved the tissue content of EPA and DHA even more than when each administered individually in animals fed HFD. Furthermore, EPA reduced the n-6/n-3 ratio EWAT of CD animals, and ameliorate this ratio in both groups of mice fed HFD that received EPA supplementation. Also, HT given alone diminished the ratio n-6/n-3 in EWAT of HFD-mice, an effect that was more pronounce after EPA and HT co-supplementation (Table 3).

### 3.3. HFD-Induced EWAT Oxidative Stress Enhancement is Normalized by EPA and HT Co-Administration

The treatments showed no effect on any of the parameters associated with oxidative stress evaluated in EWAT of CD-fed mice (Figure 1). HFD feeding induced oxidative stress in EWAT, as evidenced by the increase in markers of oxidative damage TBARS (Figure 1A), F2-isoprostanes (Figure 1B) and protein carbonyls (Figure 1C) accompanied by alteration of the glutathione status (Figure 1D–G). Supplementation with EPA alone only reduced levels of protein carbonyls (Figure 1C), without improving the rest of the parameters evaluated in EWAT of HFD animals (Figure 1A,B,D–G). When HT was given alone, TBARS and F2-isoprostanes levels were reduced (Figure 1A,B), and the content of protein carbonyls (Figure 1C) and glutathione levels (Figure 1D–G) were normalized, reaching control values. Notably, the administration of both supplements, EPA + HT, prevented oxidative damage in all parameters evaluated and deterioration of glutathione status in EWAT of mice receiving the HFD (Figure 1A–D).

The fall in the mRNA levels and the DNA binding capacity impaired of the cytoprotective transcription factor Nrf2 in EWAT of HFD-fed mice were partially recovered when animals received EPA or HT supplementation (Figure 2A,B). Likewise, the HFD-induced diminution in the expression levels of its downstream genes glutamate-cysteine ligase (GCL) (Figure 2C) and glutathione-S-transferase (GST) (Figure 2D) involved in glutathione metabolism were equally improved by EPA or HT. The supplementation with EPA plus HT in HFD mice showed an even greater effect, restoring the expression levels of Nrf2, GCL and GST to control values, and normalizing the DNA binding activity of Nrf2 (Figure 2A–D). A similar behavior was observed in the activity of antioxidant enzymes CAT (Figure 2E), SOD (Figure 2F), GR (Figure 2G) and GPX (Figure 2H) in EWAT of HFD-mice that received EPA, HT or both supplements, which were partially enhanced by EPA or HT after reduction by HFD and totally recovered by combined EPA and HT (Figure 2D–H).

### 3.4. HFD-Induced Upregulation of SREBP-1c System and Downregulation of PPAR-γ Pathway in EWAT are Partially Recovered by the Combined EPA and HT Protocol

The EWAT of mice subjected to HFD exhibit a significant increase in DNA-binding activity (Figure 3A) and mRNA levels of SREBP-1c (Figure 3B). The activation of SREBP-1c was accompanied by higher gene expression and activities of lipogenic enzymes ACC (Figure 3E,F) and FAS (Figure 3C,D) and higher activities of G6PD (Figure 3G) and ME (Figure 3H) compared to CD values. In addition, mRNA levels of enzyme LPL were decreased, whereas their activity increased in EWAT of HFD-fed mice (Figure 4D,E). The administration of EPA or HT supplied alone to HFD mice led to a partial recovery of all the lipogenic parameters evaluated in EWAT (Figure 3) including the LPL enzyme (Figure 4D,E), showing comparable results with both supplements, except for the FAS activity (Figure 3G). The effect of EPA and HT given together led to a significantly greater recovery of SREBP-1c (Figure 3A,B) and its target genes FAS, ACC, G6PD, ME (Figure 3C–H) and LPL (Figure 4D,E), although without achieving values observed in EWAT of CD mice.

The DNA-binding activity (Figure 4A) and mRNA levels of PPAR-γ (Figure 4B) and fatty acid transport protein 1 (FATP1) (Figure 4C) were decreased in EWAT of HFD-fed mice compared to the control group. Supplementation whit EPA or HT alone similarly improved the activity and mRNA levels of PPAR-γ and gene expression of FATP1. Moreover, EPA plus HT supplementation led to a significant increase in the binding activity and gene expression of PPAR-γ and in the gene expression of the FATP1 transporter compared to groups that received EPA or HT, without reaching CD values (Figure 4A–C). Mice subjected to CD with or without supplementation showed comparable results in all parameters evaluated in EWAT mentioned in Figure 3 and Figure 4.

### 3.5. HFD-Induced Activation of Transcription Factor NF-κΒ and Expression of Target Genes Associated with Inflammation in EWAT are Diminished to Values over CD by the EPA Plus HT Protocol

The DNA-binding activity and the gene expression of the transcription factor NF-κΒ were significantly increased in EWAT of mice fed with the HFD compared to those that received the CD (Figure 5A,B). In parallel, the mRNA levels of the pro-inflammatory cytokine tumor necrosis factor alpha (TNF-α) and interleukin 6 (IL-6) were increased in EWAT of HFD-fed mice (Figure 5C,D). In the HFD-fed mice that received EPA or HT, the DNA-binding activity of NF-κΒ and the mRNA levels of NF-κΒ, TNF-α and IL-6 were moderately improved, with comparable results between both groups, respectively. The group that received EPA plus HT showed a significantly greater decrease in these values in comparison to the groups supplemented with EPA or HT alone, without reaching those of the control group (Figure 5).

## 4. Discussion

Diet-induced obesity (DIO) in mice represents a model of choice for preclinical nutritional studies [33,34,35]. HFD feeding comprising 60% of the total calories as fat for 12 weeks achieved significant BW gain with concomitant liver and WAT dysfunction. In the case of the liver, HFD induced steatosis in association with n-3 LCPUFAs depletion, oxidative stress with loss of antioxidant defenses [20], a pro-inflammatory state without histological inflammation [17] and a decline in mitochondrial energy metabolism [18]. The drastic metabolic alterations induced by HFD are evidenced by the deranged levels of (i) serum-free fatty acids, the adipokines adiponectin and leptin, and the cytokines TNF-α and IL-6, which are associated with IR development [13], (ii) the development of adipocyte hypertrophy and enhancement in total tissue mass, with higher cell size and TAG content and lower cell number per g of tissue over control values and (iii) the derangement of metabolic parameters in WAT favoring oxidative stress, lipogenesis, mitochondrial dysfunction and inflammation, features that agree with a previous report [13] and resemble those elicited in the liver [17,18,20].

Metabolic alterations can be prevented or attenuated by the combination of active substances, including n-3 LCPUFAs, antioxidants and insulin sensitizers, as reported for DHA, EPA, HT, rosiglitazone and astaxantin [17,18,36,37]. In the present study, EPA and HT co-administration diminished BW gain in HFD-fed mice over CD animals, an effect that was accompanied by a partial WAT reduction and EWAT cellularity preservation similar to those observed in the separate supplementations. In agreement with these findings, replacement of dietary lipids in a HFD by 1% to 12% (*wt/wt*) of EPA plus DHA (DHA > EPA) limited the BW gain and EWAT accumulation in C57BL/6J HFD-fed mice, with similar results being found by replacing a HFD rich in ALA or linoleic acid (C18:2n-6, LA) by mixtures of EPA and DHA at different ratios [38]. The effect of EPA on obesity may be partly mediated by binding to G protein-coupled receptor 120 (GPR120) present in adipocytes and adipocyte tissue macrophages (ATMs) [39], thus regulating adipogenesis, glucose uptake, inflammation and insulin sensitivity [40]. EPA binding and activation of GPR120 promotes several effects, including (i) the interaction with PPAR-γ that favors the differentiation of pre-adipocytes into white adipocytes [40], (ii) the binding to β-arrestin2 and their interaction with transforming growth factor-β-activated kinase binding protein 1 (TAB1) that blocks NF-κΒ and c Jun N-terminal kinase (JNK) functioning, with inhibition of the low-grade inflammation due to TNF-α and IL-6 production by ATMs triggered by obesity [41] and (iii) the release of fibroblast growth factor 21 (FGF21), resulting in the promotion of brown adipose tissue (BAT) activity and WAT browning [42], with stimulation of mitochondrial energy metabolism depressed by HFD. It is important to note that EPA supplementation in HFD-fed mice upregulates the expression of PPAR-α and the micro-RNAs miR-455 and miR-129-5p in BAT [43]. These mediators enhance the expression of thermogenic markers that stimulate mitochondrial FAO and energy expenditure, pointing to BAT as a crucial anti-obesity target [43]. This contention is further supported by data showing that HT prevents visceral adipogenesis and decreased metabolic activity in BAT and WAT, induced by the exposure to fine particulate matter in mice [44]. The contribution of EPA to the induction of a healthy adipocyte phenotype may also involve its direct binding to PPAR-γ, whose expression and activity are diminished by HFD and partially rescued by EPA. Activation of PPAR-γ induces adipogenesis as an important mechanism of WAT remodeling, which is accomplished through downregulation of SREBP-1c functioning [9,39]. The HT component of the combined protocol elicited diminution of WAT lipid accumulation by regulating genes related to lipogenesis and FAO, but mainly due to the improvement of the cellular antioxidant status that preserves n-3 LCPUFAs loss by HFD [13,45]. The latter mechanism of HT action is exerted directly as a free-radical scavenger or by recovering the Nrf2 system that induces antioxidant enzymes (SOD, CAT, GPX) and those associated with glutathione repletion (GCL, GR) or utilization (GST). At the doses used in our study, the combined EPA and HT administration did not exhibit more potentiated effects on WAT morphology parameters and lipid content than those of the individual compounds. However, the parameters related to WAT metabolic dysfunction are diminished in an additive manner, considering that the specific effects exerted by EPA or HT exhibit convergence in the combined protocol.

In agreement with previous data, WAT expansion in HFD-fed mice led to a significant decrease of serum levels of adiponectin and an increase in those of leptin [13], which are correlated with IR [46] and are partially restored by EPA and HT single or combined supplementations. In this context, the administration of EPA is known to enhance the gene expression and release of adiponectin from adipocytes, either in EWAT from HFD-fed mice [47] or in lean and overweight rats [48], thus evidencing the insulin-sensitizing properties of EPA [17,47,48]. Interestingly, the administration of EPA to obese subjects elevated adiponectin levels without altering BW, suggesting direct involvement of EPA on adiponectin secretion [49]. This conclusion is also supported by EPA supplementation to HFD-fed mice showing maintenance of normal adiponectin levels without diminishing adiposity, pointing to adiponectin as a major factor in EPA-mediated prevention and reversal of IR [50]. Regarding the molecular mechanisms underlying these effects, EPA increase the expression of adiponectin through binding to PPAR-γ, promoting the expression of FGF21 and the activation of AMP-activated protein kinase (AMPK), the energy sensor that favors FAO over lipogenesis [15,42,47,51]. The EPA-induced AMPK activation by phosphorylation in 3T3-L1 adipocytes [51] is in agreement with in vivo studies showing that n-3 LCPUFAs increase the content of WAT AMPK α1 subunit and its phosphorylation status, in association with an enhancement in adiponectin levels [52,53]. In the case of leptin, the enhancement in its serum levels by HFD feeding in mice and their diminution by EPA (Table 2D) agree with studies in humans and rodents [54], suggesting that improved leptin sensitivity concomitantly with decreased production would be secondary to the anti-inflammatory effect of n-3 LCPUFAs [55]. Regarding HT, several preclinical studies reported higher circulating levels of adiponectin with lower leptin values than controls achieved by this polyphenol [13,24,45,56].

Improved expression and activity of the PPAR-γ gene by co-administration of EPA and HT also leads to improved expression of the target genes FATP1 and LPL, an effect that is additive over the individual supplementations after HFD feeding. In particular, upregulation of WAT FATP1 may contribute to the smaller healthy phenotype of adipocytes due to its FA transport function across cell membranes and its acyl-CoA synthase activity to activate FAs for further utilization [57]. LPL, however, exhibited opposite changes regarding its mRNA expression and activity, which may be related to different mechanisms regulating LPL at transcriptional and post-translational levels [13]. The small healthy adipocyte phenotype is also associated with downregulation of key lipogenic genes in EPA-HT-HFD-treated mice. These include genes coding for transcription factor SREBP-1c that controls the expression of the lipogenic enzymes ACC and FAS and the reduced nicotinamide adenine dinucleotide phosphate (NADPH) generating systems G6PD and ME that are required for FA synthesis. The anti-lipogenic effect of the EPA plus HT protocol is related to the repletion of n-3 LCPUFAs due to lower oxidative deterioration by the normoxic state achieved, which downregulates lipogenic SREBP-1c functioning [18]. This contention is supported by previous studies showing (i) diminution in the lipogenic enzyme activities in WAT of sucrose-rich diet-fed rats subjected to fish oil administration [58], (ii) the decrease in the expression of the lipogenic enzyme glycerol-3-phosphate dehydrogenase in EPA-treated 3T3-L1 adipocytes [50] and (iii) the diminished SREBP-1c functioning with prevention of DIO in adipocytes from HFD-fed mice supplemented with EPA [59]. Additionally, HFD-induced prolipogenic state in EWAT with n-3 LCPUFAs depletion are attenuated by HT supplementation, which may preserve the activity of delta-5 and delta-6 desaturases that contribute to n-3 LCPUFAs synthesis, as previously reported in the liver [24], downregulate SREBP-1c or reduce oxidative stress [13]. A major hallmark in the expansion of WAT in obesity is the enhancement in the adipocyte oxidative stress status, which underlies significant mitochondrial dysfunction. This is evidenced by the decreases in the activities of the carboxylic acid cycle regulatory enzyme citrate synthase and the respiratory chain complexes I and II [60], a condition that favors energy deprivation and mitochondrial reactive oxygen species (ROS) production, thus contributing to the attainment of a hypertrophic adipocyte phenotype [61,62]. Data presented indicate that HFD coupled to EPA plus HT supplementation totally avoided oxidative damage to biomolecules and the depletion of antioxidant systems in WAT. This finding mainly relies on the HT component of the combined protocol and may effectively contribute to the hypertrophic to the small healthy adipocyte phenotype transition observed. The reduction of the enlarged WAT in obesity achieved by EPA plus HT is in agreement with studies on the antioxidant behavior exerted by EPA, DHA and the carotenoid astaxantin, alone or in combinations in HepG2-C8 cells [37], EPA and DHA in 3T3-L1 adipocytes and Schwann cells [63,64] or in human macrophages [65], with responses exhibiting synergism. The EPA plus HT protocol also improved the pro-inflammatory status of WAT in HFD-fed mice in an additive manner. Possible mechanisms involved include: (i) disruption of the SFA inflammatory signal in Toll-like receptor 4 that activate NF-κB, (ii) PPAR-γ activation that interferes with the translocation of NF-κB to the nucleus, (iii) binding to GPR120 and prevention of NF-κB activation, (iv) decreased pro-inflammatory adipokine expression and macrophage accumulation, promoting repolarization towards the M2 anti-inflammatory phenotype by SPMs [9,66,67] and (v) preservation of WAT n-3 LCPUFAs and downregulation of NF-κB by HT [13,44].

## 5. Conclusions

HFD feeding in mice drastically impacts WAT functioning through the induction of hypertrophy and metabolic dysfunction. WAT hypertrophy involves higher cell size with TAG accumulation and lower cell number, which is related to EPA depletion that blunts PPAR-γ operation. Metabolic dysfunction includes (i) upregulation of the lipogenic SREBP-1c system, (ii) loss of mitochondrial energy metabolism and (iii) downregulation of the antioxidant Nrf2 system, leading to oxidative stress and the secondary n-3 LCPUFAs depletion and activation of the redox-sensitive pro-inflammatory NF-κB system. These effects of HFD were significantly diminished by the co-supplementation with EPA and HT additively, with resulting values being close or similar to those observed for the CD-fed animals. These observations strengthen the contention that EPA and HT co-administration is an important strategy to attenuate obesity-driven metabolic inflammation states. The data reported concerning PPAR-γ upregulation by EPA and HT co-administration may also be of importance as a possible adjuvant therapy to diminish the severity of SARS-CoV-2 infection [68], considering the anti-inflammatory response involving PPAR-γ binding to GPR120 and/or SPMs production from EPA and its product DHA, and the anti-fibrotic effect of PPAR-γ [69].

## Figures and Tables

**Figure 1 molecules-25-04433-f001:**
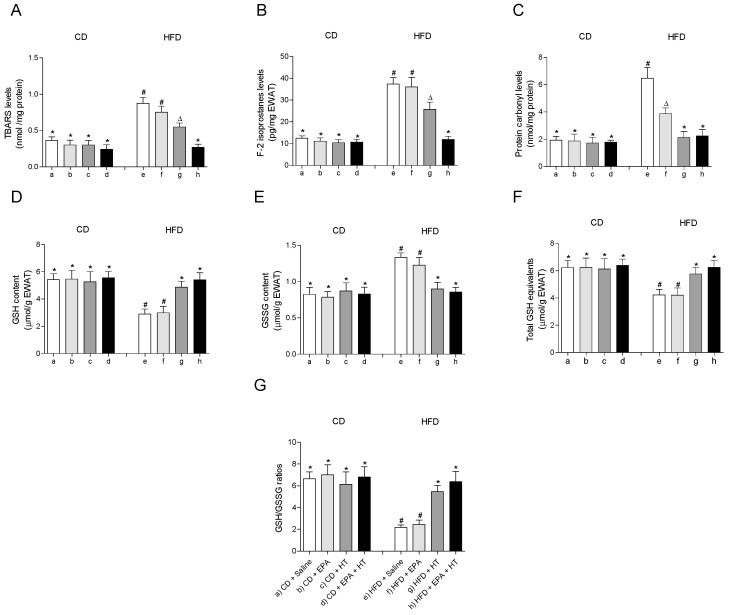
Thiobarbituric acid reactive substances (TBARS), (**A**) F-2 isoprostanes, (**B**) protein carbonyl groups, (**C**) reduced glutathione (GSH) content, (**D**) glutathione disulfide (GSSG) levels, (**E**) total GSH equivalents (GSH + 2GSSG) and (**F**) GSH/GSSG ratio (**G**) in epididymal white adipose tissue from mice subjected to control diet (CD) and high-fat diet (HFD) supplemented or not with eicosapentaenoic acid (EPA), hydroxytyrosol (HT) or EPA + HT. Values are expressed as mean ± SD. Eight to ten mice were included in each experimental group. Groups that do not share the same superscript symbol were significantly different among them, namely, (a), CD + saline; (b) CD + EPA; (c) CD + HT; (d) CD + EPA + HT; (e), HFD + saline; (f) HFD + EPA, (g) HFD + HT; and (h), HFD + EPA + HT, according to two-way analysis of variance (ANOVA) followed by the Tukey’s post-test, *p* < 0.05. (*****), does not differ from (a) group; (**#**), differ from (a) group; (∆), differ from (a) and (b) groups.

**Figure 2 molecules-25-04433-f002:**
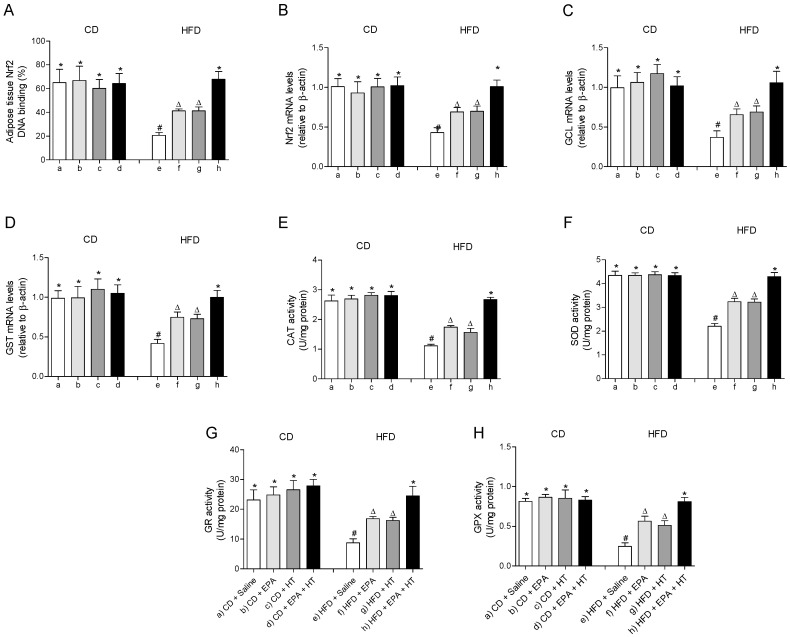
Nuclear factor E2-related factor 2 (Nrf2) DNA binding, (**A**) Nrf2 mRNA levels, (**B**) mRNA levels of glutamate-cysteine ligase (GCL) (**C**) and glutathione-S-transferase (GST), (**D**) and activity of catalase (CAT), (**E**) superoxide dismutase (SOD), (**F**) glutathione peroxidase (GPX), (**G**) and glutathione reductase (GR) (**H**) in epididymal white adipose tissue from mice subjected to control diet (CD) and high-fat diet (HFD) supplemented or not with eicosapentaenoic acid (EPA), hydroxytyrosol (HT) or EPA + HT. Values are expressed as mean ± SD. Eight to ten mice were included in each experimental group. Groups that do not share the same superscript symbol were significantly different among them, namely, (a), CD + saline; (b) CD + EPA; (c) CD + HT; (d) CD + EPA + HT; (e), HFD + saline; (f) HFD + EPA, (g) HFD + HT; and (h), HFD + EPA + HT, according to two-way analysis of variance (ANOVA) followed by the Tukey’s post-test, *p* < 0.05. (*****), does not differ from (a) group; (**#**), differ from (a) group and does not differ from (e) group; (∆), differ from (a) and (b) groups.

**Figure 3 molecules-25-04433-f003:**
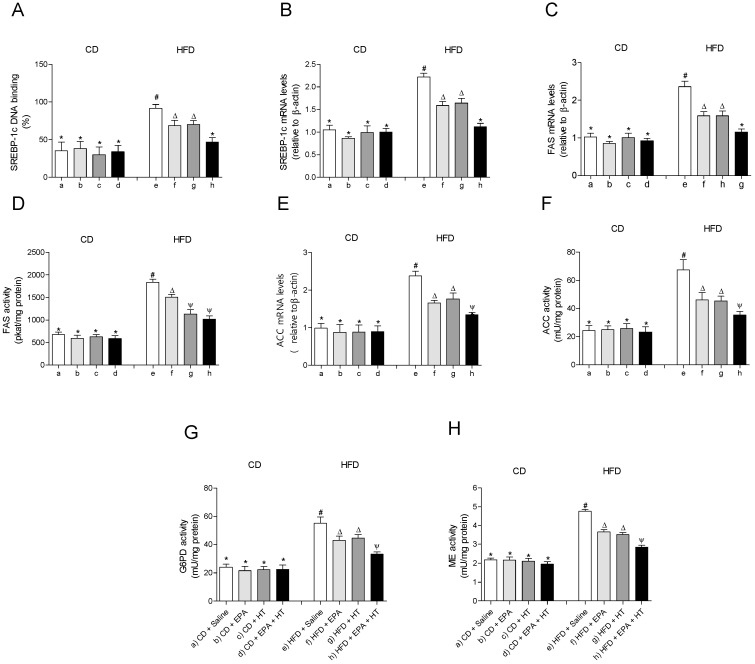
Sterol regulatory element-binding protein-1c (SREBP-1c) DNA-binding, (**A**) SREBP-1 mRNA levels, (**B**) fatty acid synthase (FAS) mRNA content, (**C**) FAS activity, (**D**) acetyl-CoA carboxylase (ACC) mRNA levels, (**E**) ACC activity, (**F**) glucose-6-phosphate dehydrogenase (G6PD) activity, (**G**) and malic enzyme (ME) activity (**H**) in epididymal white adipose tissue from mice subjected to control diet (CD) and high-fat diet (HFD) supplemented or not with eicosapentaenoic acid (EPA), hydroxytyrosol (HT) or EPA + HT. Values are expressed as mean ± SD. Eight to ten mice were included in each experimental group. Groups that do not share the same superscript symbol were significantly different among them, namely, (a), CD + saline; (b) CD + EPA; (c) CD + HT; (d) CD + EPA + HT; (e), HFD + saline; (f) HFD + EPA, (g) HFD + HT; and (h), HFD + EPA + HT, according to two-way analysis of variance (ANOVA) followed by the Tukey’s post-test, *p* < 0.05. (*****), does not differ from (a) group; (**#**), differ from (a) group; (∆), differ from (a) and (e) groups; (Ψ), differs from groups (a), (e) and those marked with the symbol ∆.

**Figure 4 molecules-25-04433-f004:**
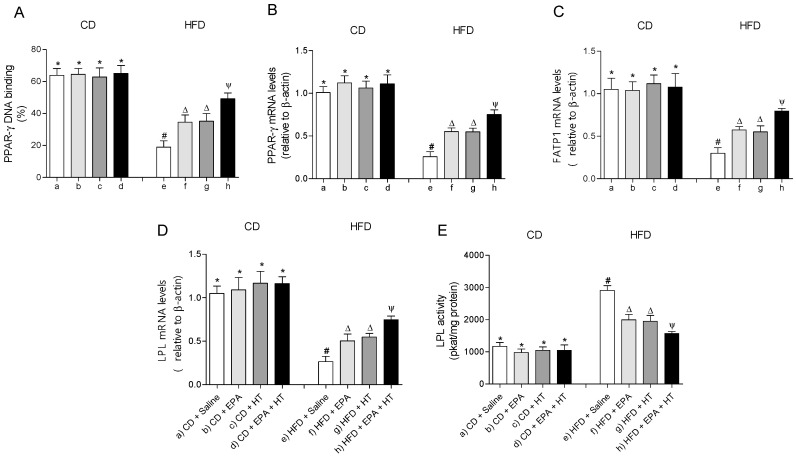
Peroxisome proliferator-activated receptor-gamma (PPAR-γ) DNA-binding, (**A**) mRNA levels of PPAR-γ, (**B**) fatty acid transport protein 1 (FATP1), (**C**) lipoprotein lipase (LPL) (**D**) and LPL activity (**E**) in epididymal white adipose tissue from mice subjected to control diet (CD) and high-fat diet (HFD) supplemented or not with eicosapentaenoic acid (EPA), hydroxytyrosol (HT) or EPA + HT. Values are expressed as mean ± SD. Eight to ten mice were included in each experimental group. Groups that do not share the same superscript symbol were significantly different among them, namely, (a), CD + saline; (b) CD + EPA; (c) CD + HT; (d) CD + EPA + HT; (e), HFD + saline; (f) HFD + EPA, (g) HFD + HT; and (h), HFD + EPA + HT, according to two-way analysis of variance (ANOVA) followed by the Tukey’s post-test, *p* < 0.05. (*****), does not differ from (a) group; (**#**), differ from (a) group; (∆), differ from (a) and (e) groups; (Ψ), differs from groups (a), (e) and those marked with the symbol ∆.

**Figure 5 molecules-25-04433-f005:**
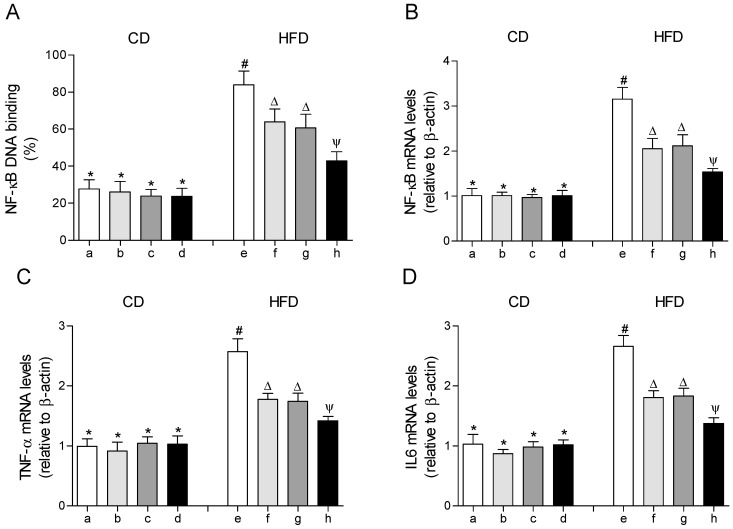
Nuclear factor erythroid 2–related factor 2 (NF-κB) DNA-binding, (**A**) mRNA levels of NF-κB, (**B**) mRNA levels of tumor necrosis factor (TNF-α), (**C**) and mRNA levels of interleukin 6 (IL-6) (**D**) in epididymal white adipose tissue from mice subjected to control diet (CD) and high-fat diet (HFD) supplemented or not with eicosapentaenoic acid (EPA), hydroxytyrosol (HT) or EPA + HT. Groups that do not share the same superscript symbol were significantly different among them, namely, (a), CD + saline; (b) CD + EPA; (c) CD + HT; (d) CD + EPA + HT; (e), HFD + saline; (f) HFD + EPA, (g) HFD + HT; and (h), HFD + EPA + HT, according to two-way analysis of variance (ANOVA) followed by the Tukey’s post-test, *p* < 0.05. (*****), does not differ from (a) group; (**#**), differ from (a) group; (∆), differ from (a) and (e) groups; (Ψ), differs from groups (a), (e) and those marked with the symbol ∆.

**Table 1 molecules-25-04433-t001:** Gene-specific primer sequences used in the study.

mRNA	Forward Primer	Reverse Primer	Melting T^o^	Product Length	Gen Bank Code
Nrf2	AAGCTTTCAACCCGAAGCAC	TTTCCGAGTCACTGAACCCA	58 °C	156	NM_0317899.2
GST	TGCAGACCAAAGCCATTCTC	ACGGTTCCTGGTTTGTTCCT	56 °C	197	NM_017013.4
GCL	ATGTGGACACCCGATGCAGTATT	TGTCTTGCTTGTAGTCAGGATGGTTT	62 °C	129	NM_012815.2
SREBP-1c	CTGGAGACATCGCAAACAAGC	ATGGTAGACAACAGCCGCATC	58 °C	277	NM_001358315.1
FAS	ATCCTGGAACGAGAACACGATCA	AGAGACGTGTCACTCCTGGACTT	59 °C	140	NM_017332.1
ACC	ACCAGGGCAAATGCATCAGT	TCGGAAAAGCATCGGGAAGT	57 °C	185	NM_022193.1
PPAR-γ	CCAGAGCATGGTGCCTTCGCT	CAGCAACCATTGGGTCAGCTC	58 °C	241	NM_011146.3
FATP1	TCCGTCTGGTCAAGGTCAAT	GAAAACGCTGTGGGCAATCT	56 °C	188	NM_011977.4
LPL	AGCCAGGATGCAACATTGGA	TTGCACCTGTATGCCTTGCT	57 °C	157	NM_008509.2
NF-κΒ	GAGGTCTCTGGGGGTACCAT	AAGGCTGCCTGGATCACTTC	58 °C	89	NM_008689.2
TNF-α	ATGGCCTCCCTCTCATCAGT	TTTGCTACGACGTGGGCTAC	58 °C	97	NM_013693.3
IL-6	TCCATCCAGTTGCCTTCTTG	TTCCACGATTTCCCAGAGAAC	56 °C	167	NM_031168.2
β-actin	ACTGCCGCATCCTCTTCCTC	CTCCTGCTTGCTGATCCACATC	58 °C	399	NM_031144.3

Sequences are listed in the 5′ → 3′ direction. Nuclear factor erythroid 2-related factor 2 (Nrf2), glutathione S-transferase (GST), glutamate-cysteine ligase (GCL), sterol regulatory element-binding protein-1c (SREBP-1c), fatty acid synthase (FAS), acetyl-CoA carboxylase (ACC), peroxisome proliferator-activated receptor-gamma (PPAR-γ), Fatty acid transport protein 1 (FATP1), lipoprotein lipase (LPL), nuclear factor kappa-light-chain-enhancer of activated B cells (NF-κB), tumor necrosis factor alpha (TNF-α), interleukin 6 (IL-6), beta actin (β-actin).

**Table 2 molecules-25-04433-t002:** General and metabolic parameters in control diet (CD) and high-fat diet (HFD)-fed mice subjected to eicosapentaenoic acid (EPA), hydroxytyrosol (HT) or EPA + HT supplementation.

	Groups							
	Control Diet (CD)				High-Fat Diet (HFD)			
	Saline (a)	EPA (b)	HT (c)	EPA + HT (d)	Saline (e)	EPA (f)	HT (g)	EPA + HT (h)
A. General parameters								
Initial body weight (g)	14.2 ± 1.1	13.9 ± 1.4	14.0 ± 1.5	14.3 ± 1.1	14.2 ± 1.3	13.8 ± 1.3	14.3 ± 1.5	13.9 ± 1.3
Final body weight (g)	37.2 ± 3.1 ^e,f,g,h^	36.9 ± 3.2 ^e,f,g,h^	39.6 ± 3.5 ^e,f,g,h^	37.4 ± 3.4 ^e,f,g,h^	53.2 ± 4.4 ^a,b,c,d,f,h^	45.9 ± 4.2 ^a,b,c,d,e^	47.1 ± 4.0 ^a,b,c,d^	44.3 ± 3.6 ^a,b,c,d,e^
B. White Adipose Tissue (WAT)								
Subcutaneous WAT (g)	0.27 ± 0.09 ^e,f,g,h^	0.26 ± 0.10 ^e,f,g,h^	0.28 ± 0.11 ^e,f,g,h^	0.26 ± 0.08 ^e,f,g,h^	0.88 ± 0.17 ^a,b,c,d,f,g,h^	0.51 ± 0.15 ^a,b,c,d,e^	0.57 ± 0.13 ^a,b,c,d,e^	0.49 ± 0.17 ^a,b,c,d,e^
Perirenal WAT (g)	0.25 ± 0.11 ^e,f,g,h^	0.23 ± 0.09 ^e,f,h,g^	0.27 ± 0.12 ^e,f,g,h^	0.22 ± 0.07 ^e,f,g,h^	0.67 ± 0.19 ^a,b,c,d^	0.49 ± 0.16 ^a,b,c,d^	0.59 ± 0.16 ^a,b,c,d^	0.44 ± 0.13 ^a,b,c,d, e^
Epididymal WAT (g)	0.53 ± 0.15 ^e,f,g,h^	0.51 ± 0.11 ^e,f,g,h^	0.55 ± 0.13 ^e,f,g,h^	0.50 ± 0.11 ^e,f,g,h^	1.74 ± 0.25 ^a,b,c,d,f,g,h^	1.15 ± 0.22 ^a,b,c,d,e^	1.21 ± 0.19 ^a,b,c,d,e^	1.17 ± 0.16 ^a,b,c,d,e^
Total weigth WAT (g)	1.12 ± 0.12 ^e,f,g,h^	1.09 ± 0.14 ^e,f,g,h^	1.15 ± 0.14 ^e,f,g,h^	1.06 ± 0.12 ^e,f,g,h^	3.34 ± 0.31 ^a,b,c,d,f,g,h^	2.21 ± 0.20 ^a,b,c,d,e^	2.40 ± 0.25 ^a,b,c,d,e^	2.26 ± 0.22 ^a,b,c,d,e^
C. Specific parameter of Epididymal WAT								
Relative weight (g/100 g body weight)	1.67 ± 0.16 ^e,f,g,h^	1.65 ± 0.13 ^e,f,g,h^	1.63 ± 0.16 ^e,f,g,h^	1.62 ± 0.14 ^e,f,g,h^	3.11 ± 0.34 ^a,b,c,d,f,g,h^	2.09 ± 0.20 ^a,b,c,d,e^	2.15 ± 0.23 ^a,b,c,d,e^	2.05 ± 0.21 ^a,b,c,d,e^
Cell volumen (pL)	259.4 ± 22.0 ^e,f,g,h^	240.4 ± 20.1 ^e,f,g,h^	248.5 ± 23.3 ^e,f,g,h^	239.4 ± 24.2 ^e,f,g,h^	498 ± 40.2 ^a,b,c,d,f,g,h^	348.4 ± 30.3 ^a,b,c,d,e^	351.1 ± 40.6 ^a,b,c,d,e^	342.1 ± 27.7 ^a,b,c,d,e^
Cell number (×10^6^/g tissue)	4.15 ± 0.37 ^e,f,g,h^	4.11 ± 0.29 ^e,f,g,h^	4.17 ± 0.32 ^e,f,g,h^	4.09 ± 0.25 ^e,f,g,h^	2.91 ± 0.21 ^a,b,c,d,f,g,h^	3.32 ± 0.28 ^a,b,c,d,e^	3.26 ± 0.21 ^a,b,c,d,e^	3.40 ± 0.33 ^a,b,c,d,e^
Triacylglicerols (nmol/cell)	0.37 ± 0.09 ^e,f,g,h^	0.35 ± 0.10 ^e,f,g,h^	0.34 ± 0.11 ^e,f,g,h^	0.31 ± 0.08 ^e,f,g,h^	0.68 ± 0.18 ^a,b,c,d,f,g,h^	0.45 ± 0.10 ^a,b,c,d,e^	0.49 ± 0.13 ^a,b,c,d,e^	0.40 ± 0.07 ^a,b,c,d,e^
D. Serum parameters								
Adiponectin (mg/L)	3.55 ± 0.21 ^e,f,g,h^	3.59 ± 0.22 ^e,f,g,h^	3.48 ± 0.24 ^e,f,g,h^	3.50 ± 0.24 ^e,f,g,h^	2.15 ± 0.14 ^a,b,c,d,f,g,h^	3.02 ± 0.14 ^a,b,c,d,e^	3.06 ± 0.17 ^a,b,c,d,e^	3.12 ± 0.15 ^a,b,c,d,e^
Leptin (mg/mL)	1.14 ± 0.08 ^e,f,g,h^	1.09 ± 0.10 ^e,f,g,h^	1.15 ± 0.11 ^e,f,g,h^	1.10 ± 0.08 ^e,f,g.h^	2.69 ± 0.29 ^a,b,c,d,f,g,h^	1.83 ± 0.20 ^a,b,c,d,e^	1.91 ± 0.26 ^a,b,c,d,e^	1.78 ± 0.17 ^a,b,c,d,e^

Values are expressed as mean ± SD; eight to ten mice were included in each experimental group. Significant differences between the groups are indicated by the letter identifying each group, namely, (a), CD + saline; (b) CD + EPA; (c) CD + HT; (d) CD + EPA + HT; (e) HFD + saline; (f) HFD + EPA, (g) HFD + HT; and (h), HFD + EPA + HT. *p* < 0.05, two-way analysis of variance (ANOVA) followed by Tukey’s post-test.

**Table 3 molecules-25-04433-t003:** Most relevant fatty acid composition of epididymal white adipose tissue (WAT) phospholipids obtained from control diet- (CD) and high-fat diet (HFD)-fed mice subjected to eicosapentaenoic acid (EPA), hydroxytyrosol (HT) or EPA plus HT supplementation.

	Fatty Acid Composition (g/100 g FAME)							
	Control Diet (CD)				High-Fat Diet (HFD)			
Fatty Acid	Saline (a)	EPA (b)	HT (c)	EPA + HT (d)	Saline (e)	EPA (f)	HT (g)	EPA + HT (h)
C16:0	36.5 ± 3.9 ^e,f,g^	33.7 ± 4.5 ^e,f,g,h^	34.9 ± 3.7 ^e,f,g,h^	34.8 ± 3.2 ^e,f,g,h^	48.7 ± 4.9 ^a,b,c,d,f,g,h^	40.8 ± 4.1 ^a,b,c,d,e^	41.5 ± 4.6 ^a,b,c,d,e^	39.8 ± 3.5 ^b,c,d,e^
C18:0	8.04 ± 1.2	8.12 ± 1.4	8.19 ± 1.5	8.16 ± 1.2	8.05 ± 1.8	8.11 ± 1.5	8.54 ± 1.3	8.14 ± 1.4
C18:1n-9	27.2 ± 2.8	26.4 ± 3.0	27.1 ± 3.1	27.4 ± 2.5	25.2 ± 3.2	27.1 ± 2.1	23.5 ± 2.4	25.9 ± 2.8
C18:2n-6 (LA)	8.12 ± 1.3	8.22 ± 1.2	8.01 ± 1.2	8.04 ± 1.5	7.87 ± 1.2	7.24 ± 1.6	7.91 ± 1.2	7.78 ± 1.3
C18:3n-3 (ALA)	2.10 ± 0.5 ^e,f^	2.21 ± 0.5 ^e,f^	1.98 ± 0.4 ^e^	1.94 ± 0.5 ^e^	0.52 ± 0.2 ^a,b,c,d,f,g,h^	1.22 ± 0.3 ^a,b,e^	1.79 ± 0.2 ^e^	1.42 ± 0.3 ^e^
C20:4n-6 (AA)	7.41 ± 0.5 ^b,d,e,f^	6.04 ± 0.3 ^a,c,e,f^	7.84 ± 0.5 ^b,d,e,f^	6.14 ± 0.3 ^a,c,e,f^	4.05 ± 0.2 ^a,b,c,d,f,g,h^	5.76 ± 0.4 ^a,c,e,g,h^	6.98 ± 0.4 ^b,d,e,f^	7.01 ± 0.5 ^b,d,e,f^
C20:5n-3 (EPA)	1.10 ± 0.2 ^b,d,e,g^	3.15 ± 0.5 ^a,c,e,f,g,h^	1.31 ± 0.3 ^b,d,e,g^	3.32 ± 0.4 ^a,c,e,f,g,h^	0.21 ± 0.01 ^a,b,c,d,f,g,h^	0.94 ± 0.2 ^b,d,e,g,h^	0.58 ± 0.1 ^a,b,c,d,e,f,h^	1.50 ± 0.3 ^b,d,e,f,g^
C22:6n-3 (DHA)	2.58 ± 0.3 ^b,d,e,g^	3.76 ± 0.5 ^a,c,e,f,g,h^	2.81 ± 0.4 ^b,d,e,g^	3.88 ± 0.4 ^a,c,e,f,g,h^	0.72 ± 0.05 ^a,b,c,d,f,g,h^	2.14 ± 0.2 ^b,d,e^	1.95 ± 0.2 ^a,b,c,d,e,h^	2.64 ± 0.3 ^b,d,e,g^
Total SFAs	47.0 ± 4.3 ^e^	43.7 ± 4.6 ^e^	45.1 ± 4.9 ^e^	44.9 ± 4.7 ^e^	58.9 ± 5.8 ^a,b,c,d^	51.8 ± 5.4	52.9 ± 5.0	50.1 ± 4.4 ^e^
Total MUFAs	29.3 ± 2.9	30.4 ± 3.1	31.4 ± 3.2	30.8 ± 2.7	27.1 ± 3.1	29.3 ± 2.5	24.7 ± 3.1	27.6 ± 3.3
Total PUFAs	23.7 ± 2.5 ^e,f^	25.9 ± 2.7 ^e,f^	23.5 ± 2.4 ^e,f^	24.3 ± 2.1 ^e,f^	14.0 ± 1.4 ^a,b,c,d,f,g,h^	18.9 ± 1.8 ^a,b,c,d,e^	22.4 ± 2.4 ^e^	22.3 ± 2.2 ^e^
Total LCPUFAs	12.9 ± 0.4 ^e,f,g,h^	13.7 ± 0.5 ^e,f,g,h^	13.1 ± 0.3 ^e,f,g,h^	14.0 ± 0.4 ^e,f,g,h^	5.82 ± 0.2 ^a,b,c,d,f,g,h^	9.76 ± 0.4 ^a,b,c,d,e,h^	10.4 ± 0.4 ^a,b,c,d,e,h^	11.7 ± 0.3 ^a,b,c,d,e,f,g^
Total n-6 LCPUFAs	8.78 ± 0.3 ^b,d,e,f,g,h^	6.14 ± 0.2 ^a,c,e,g,h^	9.04 ± 0.4 ^b,d,e,f,g,h^	6.46 ± 0.2 ^a,c,e,g,h^	4.42 ± 0.2 ^a,b,c,d,f,g,h^	6.27 ± 0.4 ^a,c,e^	7.72 ± 0.5 ^a,c,e^	7.22 ± 0.5 ^a,c,e^
Total n-3 LCPUFAs	4.12 ± 0.2 ^b,d,e,f,g^	7.56 ± 0.3 ^a,c,e,f,g,h^	4.06 ± 0.2 ^b,d,e,f,g^	7.54 ± 0.4 ^a,c,e,f,g,h^	1.40 ± 0.04 ^a,b,c,d,f,g,h^	3.49 ± 0.2 ^a,b,c,d,e,g,h^	2.68 ± 0.1 ^a,b,c,d,e,f,h^	4.48 ± 0.04 ^b,d,e,f,g^
n-6/n-3 LCPUFAs ratio	2.13 ± 0.1 ^b,d,e,f,g,h^	0.81 ± 0.0 ^a,c,e,f,g,h^	2.22 ± 0.1 ^b,d,e,f,g,h^	0.86 ± 0.03 ^a,c,e,f,g,h^	3.15 ± 0.2 ^a,b,c,d,f,g,h^	1.80 ± 0.03 ^a,b,c,d,e,g^	2.88 ± 0.2 ^a,b,c,d,e,f,h^	1.61 ± 0.1 ^a,b,c,d,e,g^

Values are expressed as mean ± standard deviation (SD). Eight to ten mice were included in each experimental group. Significant differences between the groups are indicated by the letter identifying each group, namely, (a), CD + saline; (b) CD + EPA; (c) CD + HT; (d) CD + EPA + HT; (e), HFD + saline; (f) HFD + EPA, (g) HFD + HT; and (h), HFD + EPA + HT. *p* < 0.05, two-way analysis of variance (ANOVA) followed by Tukey’s post-test. Fatty acid methyl esters (FAME). Saturated fatty acids (SFAs) correspond to C12:0, C14:0, C16:0 and C18:0. Monounsaturated fatty acids (MUFAs) correspond to C14:1n-7, C16:1n-7 and C18:1n-9. Polyunsaturated fatty acids (PUFAs) correspond to C18:2n-6, C18:3n-3, C20:4n-6, C20:5n-3, C22:5n-3 and C22:6n-3. n-6 LCPUFAs are C20:4n-6 and C22:5n-3, n-3 LCPUFAs are C20:5n-3, C22:5n-3 and C22:6n-3, n-6/n-3 LCPUFAs ratio: C20:4n-6/(C20:5n-3 + C22:5n-3 + C22:6n-3).

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
