# Peer review of "Protective Effects of Eicosapentaenoic Acid Plus Hydroxytyrosol Supplementation Against White Adipose Tissue Abnormalities in Mice Fed a High-Fat Diet"

_molecules, 2020, doi:10.3390/molecules25194433_

Round 1
Reviewer 1 Report
Well written manuscript with convincing data of the protective actions of EPA and HT in mitigating against DIO in C57BL/6J mice.
The hypothesis that co-administration of EPA and HT synergizes the inhibition of adiposity and WAT dysfunction in HFD-fed mice compared with separate supplementation with EPA or HT was well addressed by the methods chosen. The graphs and tables showed observations and good evidence to support the hypothesis.
Other than some grammatical errors scattered across the manuscript, the paper is scientifically sound.
My only question is whether some of the findings could be contributed by possible actions of EPA and HT on brown adipose tissue fatty acid oxidation and lipolysis?
Overall, a good paper that should be publishable in this journal.
Author Response
We appreciate the comments
Answers:
1.-Grammatical errors across the manuscript were corrected.
2.-Some of the findings could be contributed by possible actions of EPA and HY on brown adipose tissue fatty acid oxidation and lipolysis?
ANSWER: This aspect was mentioned in the second paragraph of the Discussion, lines 133-139, and is further emphasized by the addition of the following estatement.
“it is important to note that EPA supplementation in HFD-fed mice upregulates the expression of PPAR-alfa and the micro RNAs miR-455 and miR-129-5p in BAT [44]. These mediators enhance rhe expression of thermogenic markers that stimulate mitochondrial FAO and energy expenditure, pointing to BAT as a crucial anti-obesity target [45].
New references:
44. Pahlavani, M.; Wijayatunga, N.N.; Kalupahana, N.S.; Ramalingam, L.; Gunaratne, P.H.; Coarfa, C.; Rajapakshe, K.; Kottapalli, P.; Moustaid-Moussa, N. Transcriptic and microRNA analyses of gene meteorks regulated by eicosapentaenoic acid in brown adipose tissue of diet-induced obese mice. Biochim. Biophys. Acta Mol. Cell. Biol. Lipids. 2018, 1863, 1523-1531. doi:10.1016/j.bbalip.2018.09.004.
45. Wang, N.; Ma, Y.; Liu, Z.; Liu, L.; Yang, K.; Wei, Y.; Liu, Y.; Chen, X.; Sun, X.; Wen, D. Hydroxytyrosol prevents PM2.5-indiced adiposity and onsulin resistance by restraining oxidative stress related ti NF-κΒ pathway and modulation of gut microbiota in a murine model. Free Radic. Biol. Med. 2019, 141, 393-407. doi: 10/1016/j.freeradbiolmed.2019.07.002.
All changes introduced are indicated in yellow and the references were renumbered.
Reviewer 2 Report
The presented work is a continuation of the team's previous works on protective effects of eicosapentaenoic acid (EPA) and hydroxytyrosol (HT), individually applied or co-administered, against different abnormalities in mice fed a high-fat diet.
The presented experiments were properly planned including many parameters and proper methods. To me, the main problem lies in the manner of writing. As a whole, the manuscript seemed to be well written, however, it is difficult to reading. Some problems exist as far as abbreviations are concerned. On first sight, they were included properly. Some of them, however, were not explained at all or explained only later in the text or Tables, e.g. DHA, TAG, TG, MUFA, ACC, etc.
Because, some papers in this area were published by the Authors previously, the scientific novelty of the present results, i.e. additive effects of EPA and HT in adipose tissue abnormalities should be emphasized. Some results are not clear to me, e.g. significant differences between the groups in respective Figures. The labels in the charts were not explained in their legends. What does it mean “significant differences between the groups are indicated by the letter identifying each group”? Results presented in Tables are clear, but these in the charts are not.
What is more, the observed additive effects of EPA and HT were not discussed enough. The Authors cited a lot of papers, and in consequence, their own results were not presented clearly. An additive manner of action of EPA and HT was not fully explained.
Author Response
We appreciate these comments.
1. Grammatical errors across the manuscript were corrected.
2. All abbreviations were corrected in the manuscript.
3. The redaction of legendes were improved.
4. The discussion was improved.
5. All changes introduced are indicated in yellow and the references were renumbered.
Thank you for reviewing our manuscript.